# Treatment Resulting Changes in Volumes of High-^18^F-FDG-Uptake Adipose Tissues over Orbit and Epicardium Correlate with Treatment Response for Non-Hodgkin’s Lymphoma

**DOI:** 10.3390/ijms24032158

**Published:** 2023-01-21

**Authors:** Yu-Ming Huang, Chen-Hsi Hsieh, Shan-Ying Wang, Chin-Ho Tsao, Jehn-Chuan Lee, Yu-Jen Chen

**Affiliations:** 1Department of Radiation Oncology, Taipei Hospital, Ministry of Health and Welfare, New Taipei City 242, Taiwan; 2Department of Medicine, MacKay Medical College, New Taipei City 252, Taiwan; 3Department of Biomedical Imaging and Radiological Sciences, National Yang Ming Chiao Tung University, Taipei 112, Taiwan; 4Institute of Traditional Medicine, National Yang Ming Chiao Tung University, Taipei 112, Taiwan; 5Faculty of Medicine, School of Medicine, National Yang Ming Chiao Tung University, Taipei 112, Taiwan; 6Division of Radiation Oncology, Department of Radiology, Far Eastern Memorial Hospital, New Taipei City 220, Taiwan; 7Department of Nuclear Medicine Center, Far Eastern Memorial Hospital, New Taipei City 220, Taiwan; 8Department of Nuclear Medicine, MacKay Memorial Hospital, Taipei 104, Taiwan; 9Institute of Clinical Medicine, National Yang Ming Chiao Tung University, Taipei 112, Taiwan; 10Department of Otolaryngology, MacKay Memorial Hospital, Taipei 104, Taiwan; 11Department of Radiation Oncology, MacKay Memorial Hospital, Taipei 104, Taiwan; 12Department of Medical Research, MacKay Memorial Hospital, Taipei 104, Taiwan; 13Department of Artificial Intelligence and Medical Application, MacKay Junior College of Medicine, Nursing, and Management, New Taipei City 252, Taiwan; 14Department of Medical Research, China Medical University Hospital, Taichung 404, Taiwan

**Keywords:** brown adipose tissue, epicardium, orbit, non-Hodgkin’s lymphoma, R-CHOP, PET/CT

## Abstract

Background: A regimen of rituximab, cyclophosphamide, doxorubicin, vincristine, and prednisone (R-CHOP) is the standard treatment for non-Hodgkin’s lymphoma. Brown adipose tissue possesses anti-cancer potential. This study aimed to explore practical biomarkers for non-Hodgkin’s lymphoma by analyzing the metabolic activity of adipose tissue. Methods: Twenty patients who received R-CHOP for non-Hodgkin’s lymphoma were reviewed. Positron emission tomography/computed tomography (PET/CT) images, lactate dehydrogenase (LDH) levels, and body mass index (BMI) before and after treatment were collected. Regions with a high standardized uptake value (SUV) in epicardial and orbital adipose tissue were selected and analyzed by a PET/CT viewer. The initial measurements and changes in the high SUV of epicardial and orbital adipose tissues, LDH levels, and BMI of treatment responders and non-responders, and complete and partial responders, were compared. Results: The volumes of high-SUV epicardial and orbital adipose tissues significantly increased in responders after R-CHOP (*p* = 0.03 and 0.002, respectively). There were significant differences between changes in the high-SUV volumes of epicardial and orbital adipose tissues (*p* = 0.03 and 0.001, respectively) and LDH levels (*p* = 0.03) between responders and non-responders. The changes in high-SUV epicardial adipose tissue volumes were greater among complete responders than partial responders (*p* = 0.04). Poorer treatment responses were observed in patients with lower high-SUV epicardial adipose tissue volumes and higher LDH levels after R-CHOP (*p* = 0.03 and 0.03, respectively). Conclusions: The preliminary results of greater changes in high-SUV epicardial and orbital adipose tissue volumes among responders indicate that brown adipose tissue could be considered a favorable prognostic biomarker.

## 1. Introduction

Non-Hodgkin’s lymphoma (NHL) is the tenth most common cancer and the eleventh leading cause of cancer-related deaths worldwide [1]. The standardized incidence of NHL is 19.5% per 100,000 people in a year. The 5-year survival rate is 72%. The median age of patients at diagnosis is 67 years [2]. NHL is a heterogeneous group of lymphoproliferative disorders that originates from B- and T-lymphocytes. The major subtypes of NHL include diffuse large B-cell lymphoma (DLBCL), chronic lymphocytic leukemia/small lymphocytic lymphoma (CLL/SLL), and follicular lymphoma (FL), with a prevalence of 32%, 19%, and 17%, respectively [3,4].

Rituximab is a chimeric mouse/human monoclonal antibody therapeutic agent with binding specificity to CD20. CD20 is found on B cells, which are cancerous in NHL [5]. Rituximab-based chemoimmunotherapy is widely used in patients with previously untreated CD20-positive FL and DLBCL. A regimen including Rituximab, cyclophosphamide, doxorubicin, vincristine, and prednisone (R-CHOP) may be the preferred treatment for these patients. The overall response rate to R-CHOP in FL is over 90%, and the 10-year survival rate without progression is approximately 50% [4,6]. For DLBCL, R-CHOP is the mainstay of therapy and is associated with superior long-term outcomes compared to those treated with chemotherapy alone [4,6].

There is a strong relationship between the response rates of R-CHOP and the survival rates for NHL [7]. The survival rates for non-responders are inferior compared to those of responders. However, there are no established prognostic markers for identifying non-responders in NHL. If non-responders are predictively identified, timely changes in diagnostic and adaptive second-line treatment may be considered.

White adipose tissue (WAT) is the most abundant tissue in the human body. It is an energy source that can store and release energy in the form of lipids [8]. Orbital adipose tissue (OAT) contains a steady volume of WAT, which fills the spaces between the eye bulb, extraocular muscles, vessels, and nerves [9]. In contrast, brown adipose tissue (BAT) consists of iron-rich mitochondria and several lipid droplets in the cytoplasm. It plays an important role in uncoupled respiration via the uncoupling proteins (UCPs) in the mitochondria [10]. Mitochondrial uncoupling protein 1 (UCP1) is responsible for non-shivering thermogenesis in BAT and is related to energy expenditure and weight loss [11]. BAT promotes an anti-inflammatory phenotype and decreases insulin resistance [12,13]; it also possesses anti-obesity and anti-cancer potential [11,14,15,16,17]. However, the association of BAT activation with cancer progression, which is evident in rodent models, has not been verified in clinical studies [18,19]. Functional BAT presents at limited sites, such as in the interscapular, subscapular, cervical, epicardial, paravertebral, and perirenal tissues [20]. Epicardial adipose tissue (EAT) constitutes stable BAT. It is found in a limited space between the myocardium and the visceral layer of the pericardium [21]. WAT, when exposed to certain stimuli (e.g., cold, exercise, or adrenergic receptor activation), may undergo morphologic and functional changes to transform into BAT [15]. This process is known as browning or beiging [22]. Beige adipose tissue (bAT) resembles BAT, which has iron-rich mitochondria and contains UCPs. However, bAT can lose UCP1 expression after the impact of a stimulus, demonstrating that beiging is reversible [15,23]. In recent years, several imaging strategies have been used in studies related to BAT [24]. To date, positron emission tomography/computed tomography (PET/CT) using the tracer ^18^F-fluorodeoxyglucose (^18^F-FDG) has been the most commonly used strategy [24,25]. It demonstrates information about the distribution and metabolic activity of BAT [26,27,28].

Multiple clinical prognostic indices have been developed for risk stratification in NHL, such as the international prognostic index (IPI) for DLBCL [29], and the follicular lymphoma international prognostic index (FLIPI) for FL [30]. Serum lactate dehydrogenase (LDH) levels can be used for the prognostic evaluation of NHL [31]. Several prospective studies have investigated immunological biomarkers for NHL, including interleukin (IL)-6, IL-10, tumor necrosis factor (TNF)-α, CXCL13, soluble CD23 (sCD23), sCD27, and sCD30 [32,33]. Some genomic and molecular biomarkers of NHL have also been investigated, including MYC, BCL2, BCL6, and TP53 [32]. PET/CT is a sensitive and specific non-invasive imaging modality recommended for staging and restaging NHL. ^18^F-FDG PET/CT for treatment monitoring in NHL is established in current clinical practice and is not time- or labor-consuming [34].

This study aims to qualify and quantify metabolically active adipose tissue using PET/CT scans and verify this unique tissue as a prognostic biomarker for patients with NHL treated with R-CHOP.

## 2. Results

### 2.1. Patient Characteristics

Table 1 summarizes the baseline characteristics and immunohistochemical (IHC) profiles of the 20 patients. Of the 20 patients, complete response (CR) was noted in 10 patients, partial response (PR) in 7 patients, and progressive disease (PD) in 3 patients according to the post-treatment PET/CT scans. IHC analysis was performed on lymph node tissues removed during biopsy. All patients were CD20-positive in IHC profiles.

### 2.2. Pre- vs. Post-Treatment SUV-H EAT and OAT Volume, Serum LDH Levels, and BMI

Pre- and post-treatment PET/CT images were compared (Figure 1). The SUV-H EAT (high standardized uptake value epicardial adipose tissue) volume significantly increased from 6.0 ± 1.8 mL to 20.8 ± 6.4 mL after treatment in responders (*p* = 0.03) but decreased from 15.4 ± 6.5 mL to 6.8 ± 4.5 mL in non-responders (*p* = 0.32). The SUV-H OAT (high standardized uptake value orbital adipose tissue) volume significantly increased from 3.4 ± 0.7 mL to 6.5 ± 1.0 mL in responders (*p* = 0.002) but decreased from 3.3 ± 1.8 mL to 2.3 ± 1.8 mL in non-responders (*p* = 0.19). LDH levels decreased from 277.9 ± 50.3 U/L to 266.6 ± 49.3 U/L in responders (*p* = 0.86) but increased from 377.7 ± 97.2 U/L to 776.3 ± 232.7 U/L in non-responders (*p* = 0.11). Body mass index (BMI) decreased from 24.7 ± 1.0 kg/m^2^ to 24.0 ± 1.2 kg/m^2^ in responders (*p* = 0.30) and from 24.9 ± 3.1 kg/m^2^ to 22.6 ± 2.5 kg/m^2^ in non-responders (*p* = 0.29) (Figure 2). Each individual data point was plotted in Appendix A to illustrate the patient-specific local changes in the SUV-H EAT and SUV-H OAT volumes from pre- to post-treatment.

The pre- and post-treatment SUV-H EAT and SUV-H OAT volumes, LDH levels, and BMI in complete and partial responders were also compared. Among complete responders, the SUV-H EAT and SUV-H OAT volumes significantly increased from 6.3 ± 2.0 mL to 30.5 ± 9.7 mL and 4.7 ± 0.9 mL to 8.2 ± 1.1 mL, respectively, after treatment (*p* = 0.04 and 0.002, respectively). There were no significant differences in pre- and post-treatment SUV-H EAT and SUV-H OAT volumes for partial responders (*p* = 0.27 and 0.19, respectively) (Figure 3). Each individual data point was plotted in Appendix A to illustrate the patient-specific local changes in SUV-H EAT and SUV-H OAT volumes from pre- to post-treatment between complete and partial responders.

### 2.3. Responders vs. Non-Responders

The differences in delta SUV-H EAT and SUV-H OAT volumes, LDH levels, and BMI between responders and non-responders were examined using two-sample *t*-tests (Table 2). The delta SUV-H EAT volume was 14.8 ± 6.5 mL in responders compared to −8.6 ± 6.5 mL in non-responders (*p* = 0.03). The delta SUV-H OAT volume was 3.1 ± 0.8 mL in responders compared to −1.0 ± 0.4 mL in non-responders (*p* = 0.001). The delta LDH level was −11.4 ± 66.5 U/L in responders compared to 398.7 ± 151.2 U/L in non-responders (*p* = 0.03). The delta BMI was −0.7 ± 0.7 kg/m^2^ in responders compared to −2.3 ± 1.6 kg/m^2^ in non-responders (*p* = 0.36). There were statistically significant differences in the delta SUV-H EAT and SUV-H OAT volumes and serum LDH levels between responders and non-responders (Figure 4); however, there were no significant differences in initial SUV-H EAT and SUV-H OAT volumes, LDH levels, or BMI between responders and non-responders (Figure 5).

### 2.4. Complete Responders vs. Partial Responders

The results of two-sample *t*-tests for delta SUV-H EAT and SUV-H OAT volume, LDH levels, and BMI in complete and partial responders are summarized in Table 3. Delta SUV-H EAT volume was 24.2 ± 10.1 mL in complete responders and 1.3 ± 1.1 mL in partial responders (*p* = 0.04). There were no statistically significant differences in the delta SUV-H OAT volume, serum LDH levels, or delta BMI between complete and partial responders (Figure 6). There were no significant differences in initial SUV-H EAT volume, LDH levels, or BMI between complete and partial responders. Significantly greater initial SUV-H OAT volumes were observed in complete responders compared to partial responders (*p* = 0.02) (Figure 7).

### 2.5. Correlation Analysis

Pearson’s correlation analyses were performed to analyze the correlations among delta and initial SUV-H EAT, SUV-H OAT volumes, LDH levels, and BMI, respectively. The correlation coefficients (*r*) are shown in Table 4 and Table 5. A strong negative correlation between the delta SUV-H OAT volume and delta LDH levels was detected (*r* = −0.70, *p* = 0.002), and a significant positive correlation between delta SUV-H EAT and OAT volume was noted (*r* = 0.50, *p* = 0.03). A significant negative correlation between the initial SUV-H OAT volume and initial LDH levels was observed (*r* = −0.55, *p* = 0.02). No other significant correlations were shown in the correlation analyses.

### 2.6. Logistic Regression Analysis

Logistic regression analyses were performed to analyze the possible indicators of treatment responders, and the results are summarized in Table 6. Among these indicators, delta SUV-H EAT volume and delta LDH levels were significantly correlated with the treatment responses [odds ratio (OR): 32.00, 95% confidence interval (CI) 1.39–737.46, *p* = 0.03; OR: 32.00, 95% CI 1.39–737.46, *p* = 0.03, respectively], which suggested that less delta SUV-H EAT volume and higher delta LDH levels after R-CHOP were associated with poorer treatment responses. No other significant correlations were shown in the logistic regression analyses.

## 3. Discussion

In this study of NHL patients, the volume of SUV-H EAT and SUV-H OAT significantly increased in responders after four to eight cycles of R-CHOP. The differences in the delta SUV-H EAT and SUV-H OAT volumes and LDH levels between responders and non-responders were statistically significant. Greater delta SUV-H EAT and OAT volumes were observed in responders, whereas greater delta LDH levels were noted in non-responders. There was a significantly greater delta SUV-H EAT volume among complete responders compared to partial responders. Logistic regression analyses indicated that higher delta SUV-H EAT volumes and lower delta LDH levels were associated with better treatment responses.

The PET/CT scan has been an important diagnostic tool in cancer staging and restaging. ^18^F-FDG localizes at regions with increased metabolic activity. However, both physiologic and pathologic processes, such as inflammation, were associated with increased ^18^F-FDG uptake [35]. Human BAT has been commonly assessed through ^18^F-FDG PET/CT using several quantification criteria. Uniform criteria, named the Brown Adipose Reporting Criteria in Imaging STudies (BARCIST 1.0), were published and recommended in 2016 [36,37]. Tissues with relatively but not significantly high SUV, which have been regarded as indicating false-positive uptake, may be considered to be BAT depending on the location and characteristics of adipose tissue in the CT scan. EAT consists of abundant, small UCP1-expressing adipocytes in a limited space and is scarcely involved in cancer [21]. OAT is a type of WAT but differs from the other WAT of the body due to its embryonic origin, functions, and structure. OAT forms smaller lobes and has larger volumes of collagen, endothelial cells, and mast cells than subcutaneous fat [9]. Malignant orbital tumors are unusual. This study did not evaluate the overall BAT status of all patients because the metabolic activity of BAT in PET/CT is easily affected by cancer and inflammation. EAT and OAT share similarities, in that are both rarely affected by malignancies and are remote from sites of cancer treatments. Therefore, we choose EAT and OAT for evaluation.

The adipose tissue with high SUV may indicate BAT or bAT. The volumes of SUV-H EAT and SUV-H OAT significantly increased in responders, and delta SUV-H EAT and SUV-H OAT volumes were significantly greater in responders than in non-responders. These findings imply that browning after R-CHOP and a higher volume of induced bAT portend a good treatment response. The prognostic power of LDH was verified by the significantly greater delta LDH levels present in non-responders compared to responders. A strong negative correlation was found between the delta SUV-H OAT volume and delta serum LDH levels. These findings may suggest that browning in EAT and OAT are strongly associated with good response and prognosis in patients with NHL after R-CHOP.

EAT and OAT might play a role in the immune system. We previously demonstrated that EAT may serve as a biomarker of survival outcomes in patients with esophageal cancer receiving neoadjuvant chemoradiation therapy [38]. EAT is composed of adipocytes, nerve cells, inflammatory cells (mainly macrophages and mast cells), stromal cells, vascular cells, and immune cells [21,39], and functions as BAT with the expression of UCP1, brown adipocyte differentiation transcription factor PR-domain-missing 16 and peroxisome-proliferator-activated receptor γ co-activator-1α [40]. In patients with coronary artery disease, the local expression of chemokine (monocyte chemotactic protein 1 (MCP1)) and inflammatory cytokines (IL-1β, IL-6, and TNF-α) was observed with significant changes in MCP1, IL-1β, IL-6, TNF-α mRNA, and protein in the epicardial adipose stores [41,42]. Philipp et al. have developed a mouse model to study Graves’ orbitopathy (GO) and found early infiltration of macrophages in the orbital region, the induction of anti-thyroid stimulating hormone receptor antibodies, the aggregation of CD8+ T cells, and BAT increase during GO onset [43]. R-CHOP may influence EAT and OAT to induce local or systemic inflammation or immune responses, which could be related to treatment responses for NHL.

In this study, the delta volumes of SUV-H EAT and SUV-H OAT, serum LDH levels, and BMI before and after R-CHOP were recorded. The bias from individualized variation was reduced by comparing changes at more than one time point. There were significantly greater delta SUV-H EAT and OAT volumes, decreased delta serum LDH levels observed among responders compared to non-responders, and greater delta SUV-H EAT volumes detected among complete responders compared to partial responders. However, there were no statistically significant differences in delta serum LDH levels between complete and partial responders. This may imply that SUV-H EAT volume has better prognostic power than serum LDH levels in differentiating the treatment responses to R-CHOP for patients with NHL.

There were several limitations to our study. First, a causal relationship between browning and good treatment response to R-CHOP in patients with NHL (implied by greater delta volumes of SUV-H EAT and SUV-H OAT and lower delta serum LDH levels in responders compared to non-responders) cannot be determined through this method and may require further investigation. Second, the number of patients in our retrospective study with PET/CT and serum LDH profiles was too small to draw a firm conclusion. This fact could be generating some bias in the results. For example, in some cases, such as risk stratification, there were no non-responders in each of the risk groups. Our preliminary findings of greater changes in SUV-H EAT and OAT volumes among responders have provided critical information for conducting further prospective studies with more enrolled patients. Third, there was a lack of standardized methods for measuring the volume of functional BAT and bAT despite how well PET/CT was able to detect the metabolically active tissues. A precise and universally accepted standardized protocol would be required for a large-scale study. Fourth, the SUV of BAT was relatively but not significantly high. If the BAT was collocated with vigorous cancer cells or in regions of severe inflammation, artifacts might be recorded as BAT. Fifth, the impact of CHOP should be examined. Finally, other possible etiologies that caused SUV-H adipose tissue should be examined in future studies. In the present study, greater delta SUV-H EAT and OAT volumes were detected among responders compared to non-responders, and the SUV-H EAT and OAT volume significantly increased in responders. These results imply that the results may be applied to responders. Because only three non-responders were included in this study, the present results, obtained using this small sample size, are not viable for drawing conclusions among non-responders.

This investigation is the first study in which R-CHOP-induced browning in EAT and OAT with prognostic effects in NHL has been verified by PET/CT scans. Our data show that R-CHOP moderately impacts the SUV-H EAT and SUV-H OAT volumes and serum LDH levels of patients with NHL. These novel findings provide critical information for prognostication. The development of a new imaging biomarker may be feasible. Clinical studies and experimental animal models are needed for further validation.

## 4. Materials and Methods

All patients included in our study were over 18 years of age, with newly diagnosed, biopsy-proven NHL. The patients were required to have a World Health Organization performance status of 0–2. Twenty patients, diagnosed between February 2017 and July 2020, who received four to eight cycles of R-CHOP therapy at our institution, were retrospectively reviewed. Pre- and post-treatment PET/CT scans, serum LDH levels, and BMI values were collected for analysis. Medical records were reviewed, and clinical information, including age, performance status, LDH level, stage, hemoglobin, and the number of nodal and extranodal areas involved, were collected for risk stratification with prognostic indices. FLIPI and IPI were used for FL and DLBCL, respectively. The IHC profiles in biopsies were also recorded.

A whole body PET/CT scan (GE Discovery, GE Healthcare, Milwaukee, WI, USA) was performed after blood glucose measurement, with a constant temperature of 22 °C regardless of the season. Patients were asked to fast for at least 6 h before the examination, and ^18^F-FDG was injected only if blood glucose was less than 200 mg/dL. Then, the patients rested during the uptake time for 60 min in an air-conditioned waiting room with a constant temperature of 22 °C. CT was obtained in the mid-expiratory phase without intravenous contrast (slice thickness, 5 mm; tube voltage, 120 kVp; tube current, 40 mAs; field of view, 50 cm), and PET images were reconstructed. The CT used in this study was examined according to the American Association of Physicists in Medicine (AAPM) and American College of Radiology (ACR) guidelines (AAPM report #74 and #96 and ACR CT QC manual), and standard quality assurance measures were performed. All included patients had no metabolic diseases, such as hypertension or diabetes mellitus. No patients received therapies related to beta-adrenergic receptors. No cancer cachexia was observed in these patients, and no significant BMI changes were reported during the treatment or follow-up periods. The clinical characteristics of each patient, including gender, age, and BMI, were recorded in Table 1.

Lugano classification criteria were introduced for response assessment using PET/CT scans according to a 5-point scale. A score of 1 meant no abnormal ^18^F-FDG uptake, while a score of 2 indicated uptake less than that by the mediastinum. A score of 3 represented uptake greater than that by the mediastinum but less than that by the liver, while scores of 4 and 5 denoted uptake greater than that by the liver and by the liver with new sites of disease, respectively. Scores of 1 to 3 were widely considered to be PET negative and a CR. A score of 4 when restaging could be considered a PR if the ^18^F-FDG uptake had declined after initial staging, while a score of 5 stood for PD [44]. Patients with CR or PR were regarded as responders, whereas patients with PD were regarded as non-responders.

Qualitative and quantitative evaluations of adipose tissues with high SUV and specific Hounsfield unit (HU) were performed on PET/CT images. According to the recommendation of BARCIST 1.0, typical HU values for adipose tissues were −190 to −10 HU, and metabolically active adipose tissue had a threshold of SUV ≥ 1.5. The regions of adipose tissue with SUV ≥ 1.5 and HU between −190 and −10 were defined as high SUV adipose tissue in this study [36]. SUV-H EAT and OAT were selected to evaluate the influence of R-CHOP on metabolically active adipose tissues in patients with NHL. Analyses were performed using a radiation therapy planning system (Eclipse Treatment Planning System v.13, Varian Medical Systems Inc., Palo Alto, CA, USA) with PET/CT viewer (Figure 8).

Statistical analysis was performed using SigmaPlot version 12.0 (Systat Software, Inc., Point Richmond, CA, USA). Numerical data were expressed as mean ± standard deviation. Paired *t*-tests were applied for the volumes of SUV-H EAT, OAT, serum LDH levels, and BMI, which were measured before and after treatment using image analyses and medical records. Two-sample *t*-tests were used to test the differences between initial and delta SUV-H EAT and SUV-H OAT volumes, LDH levels, and BMI, between responders and non-responders, as well as between complete and partial responders. Delta represented the difference between the two values. Pearson’s correlation analyses were performed to analyze the correlations among SUV-H EAT, SUV-H OAT volume, LDH levels, and BMI, and correlation coefficients (*r*) were recorded. Logistic regression analyses were used to estimate the ORs with CIs of 95% for possible indicators of responders. Differences were considered significant if *p* < 0.05 in a two-tailed test.

## 5. Conclusions

Greater changes in SUV-H EAT and SUV-H OAT volume, which may represent more browning in EAT and OAT after R-CHOP therapy in responders than in non-responders, indicate that BAT could be regarded as a favorable prognostic biomarker. This observation is supported by PET/CT scans and serum LDH profiles. This may imply that patients with elevated SUV-H EAT and SUV-H OAT volumes will respond to R-CHOP, but this hypothesis requires more investigation. The study results may be applied to responders, but the clinical application to non-responders needs further examination. However, considering the small number of cases reviewed, a larger prospective study is required for future work. Further clinical investigations and experimental animal models to validate our findings are warranted.

## Figures and Tables

**Figure 1 ijms-24-02158-f001:**
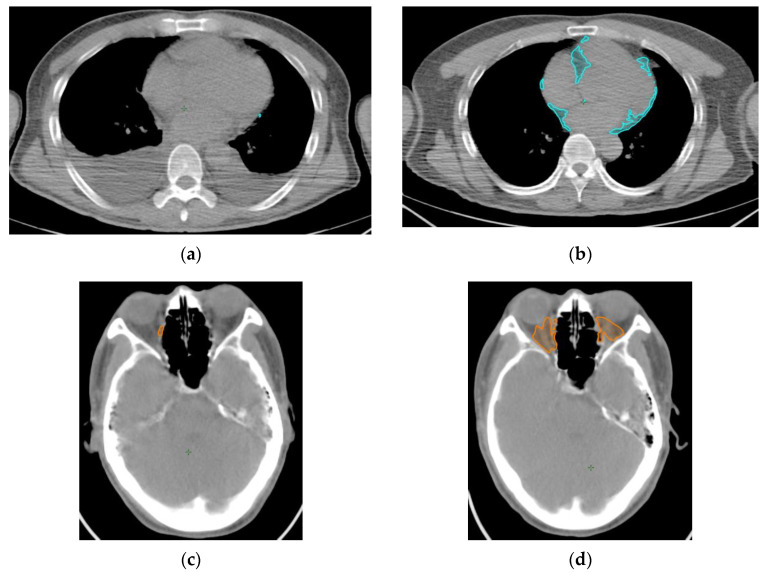
Pre- and post-treatment SUV-H EAT and SUV-H OAT for patients with different treatment responses, demonstrated by PET/CT viewer. (**a**) Pre-treatment and (**b**) post-treatment SUV-H EAT are shown in blue for a patient with CR. (**c**) Pre-treatment and (**d**) post-treatment SUV-H OAT are presented in orange for the same patient with CR. (**e**) Pre-treatment and (**f**) post-treatment SUV-H EAT are shown in blue for a patient with PD. (**g**) Pre-treatment and (**h**) post-treatment SUV-H OAT are presented in orange for the same patient with PD. SUV-H EAT, high standardized uptake value epicardial adipose tissue; SUV-H OAT, high standardized uptake value orbital adipose tissue; PET/CT, positron emission tomography/computed tomography; CR, complete response; PD, progressive disease.

**Figure 2 ijms-24-02158-f002:**
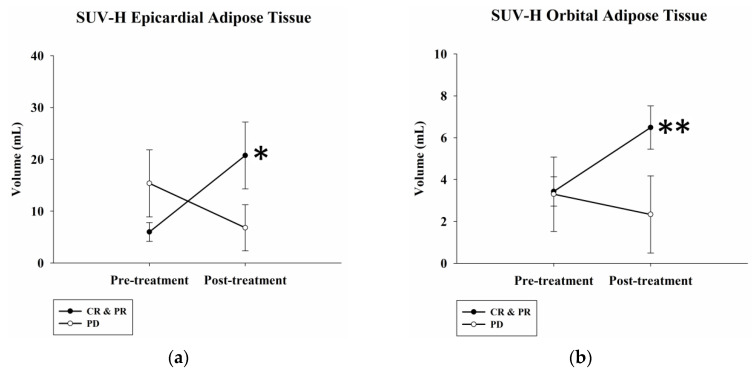
Pre- and post-treatment SUV-H EAT, OAT volume, serum LDH levels, and BMI, according to different treatment responses. (**a**) The SUV-H EAT volume significantly increased in responders (*p* = 0.03) but decreased in non-responders (*p* = 0.32). (**b**) The SUV-H OAT volume significantly increased in responders (*p* = 0.002) but decreased in non-responders (*p* = 0.19). (**c**,**d**) There were no significant differences between pre- and post-treatment serum LDH levels or BMI among responders versus non-responders. * Differences are significant at the 0.05 level (two-tailed). ** Differences are significant at the 0.01 level (two-tailed). SUV-H EAT, high standardized uptake value epicardial adipose tissue; SUV-H OAT, high standardized uptake value orbital adipose tissue; LDH, lactate dehydrogenase; BMI, body mass index; CR, complete response; PR, partial response; PD, progressive disease.

**Figure 3 ijms-24-02158-f003:**
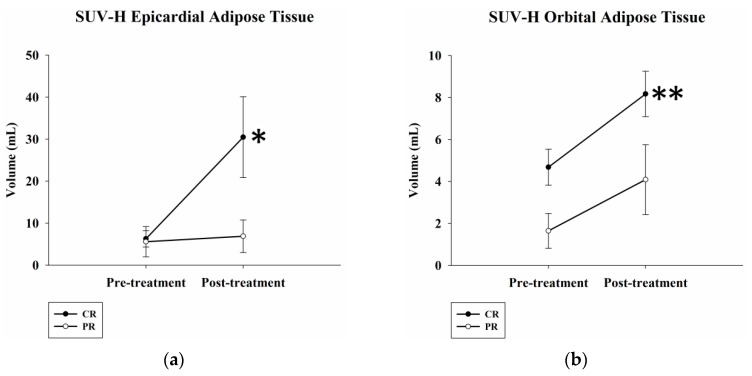
Pre- and post-treatment SUV-H EAT, SUV-H OAT volumes, serum LDH levels, and BMI in complete and partial responders. (**a**,**b**) The SUV-H EAT and OAT volume significantly increased in complete responders (*p* = 0.04 and 0.002, respectively). There were no significant differences in pre- and post-treatment SUV-H EAT and OAT volumes in partial responders (*p* = 0.27 and 0.19, respectively). (**c**,**d**) No significant differences were found in pre- and post-treatment serum LDH levels and BMI in complete and partial responders. * Differences are significant at the 0.05 level (two-tailed). ** Differences are significant at the 0.01 level (two-tailed). SUV-H EAT, high standardized uptake value epicardial adipose tissue; SUV-H OAT, high standardized uptake value orbital adipose tissue; LDH, lactate dehydrogenase; BMI, body mass index; CR, complete response; PR, partial response.

**Figure 4 ijms-24-02158-f004:**
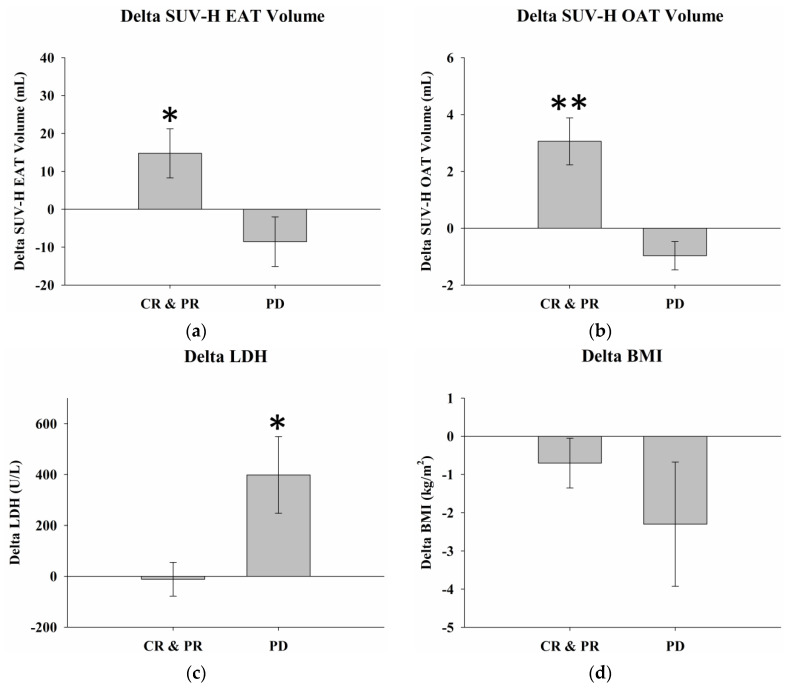
Delta SUV-H EAT, OAT volumes, serum LDH levels, and BMI with different treatment responses. (**a**) Greater delta SUV-H EAT volume was observed in responders than in non-responders (*p* = 0.03). (**b**) There was significantly more delta SUV-H OAT volume detected in responders than in non-responders (*p* = 0.001). (**c**) A significantly higher delta LDH level was noted in non-responders than in responders (*p* = 0.03). (**d**) There was no significant difference in delta BMI between responders and non-responders (*p* = 0.36). * Differences are significant at the 0.05 level (two-tailed). ** Differences are significant at the 0.01 level (two-tailed). SUV-H EAT, high standardized uptake value epicardial adipose tissue; SUV-H OAT, high standardized uptake value orbital adipose tissue; LDH, lactate dehydrogenase; BMI, body mass index; CR, complete response; PR, partial response; PD, progressive disease.

**Figure 5 ijms-24-02158-f005:**
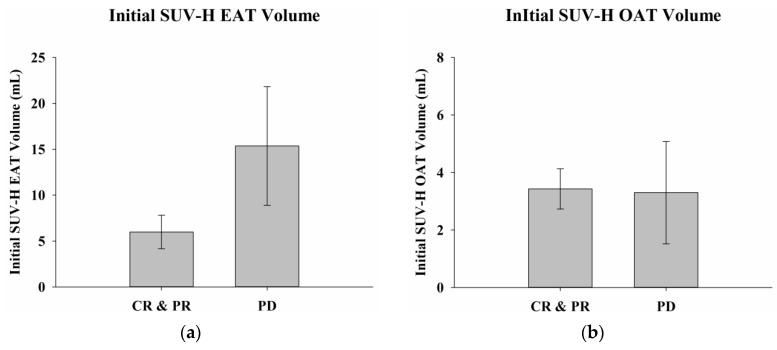
Initial SUV-H EAT and SUV-H OAT volumes, serum LDH levels, and BMI with different treatment responses. There were no statistically significant differences in (**a**) SUV-H EAT volume, (**b**) SUV-H OAT volume, (**c**) serum LDH levels, or (**d**) BMI between responders and non-responders before treatment, with *p* values of 0.08, 0.94, 0.43, and 0.95, respectively. SUV-H EAT, high standardized uptake value epicardial adipose tissue; SUV-H OAT, high standardized uptake value orbital adipose tissue; LDH, lactate dehydrogenase; BMI, body mass index; CR, complete response; PR, partial response; PD, progressive disease.

**Figure 6 ijms-24-02158-f006:**
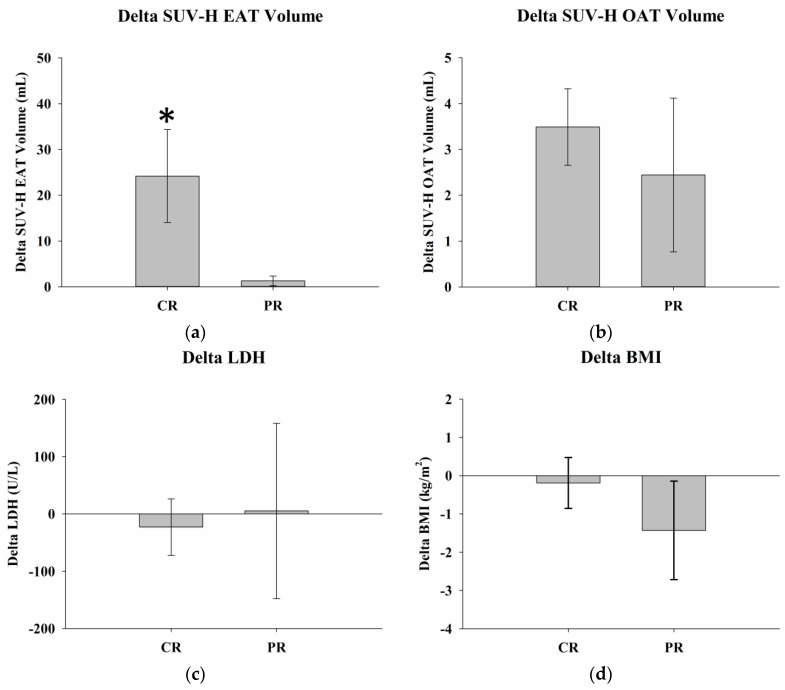
Delta SUV-H EAT and SUV-H OAT volumes, serum LDH levels, and BMI in complete and partial responders. (**a**) Greater delta SUV-H EAT volume was observed in complete responders than in partial responders (*p* = 0.04). (**b**–**d**) There were no statistically significant differences in the delta SUV-H OAT volume, serum LDH levels, or BMI between complete and partial responders, with *p* values of 0.55, 0.84, and 0.37, respectively. * Differences are significant at the 0.05 level (two-tailed). SUV-H EAT, high standardized uptake value epicardial adipose tissue; SUV-H OAT, high standardized uptake value orbital adipose tissue; LDH, lactate dehydrogenase; BMI, body mass index; CR, complete response; PR, partial response.

**Figure 7 ijms-24-02158-f007:**
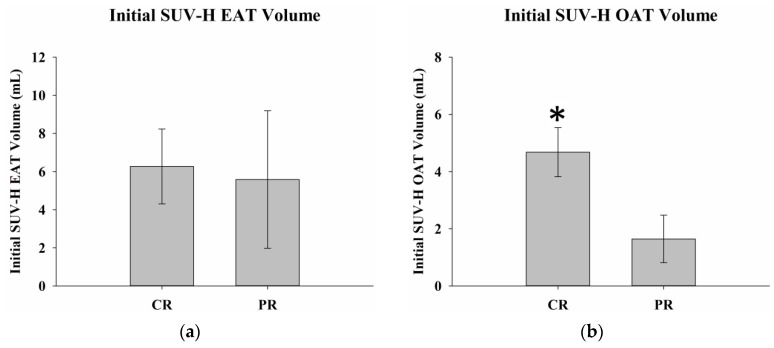
Initial SUV-H EAT and SUV-H OAT volumes, serum LDH levels, and BMI in complete and partial responders. There were no significant differences in (**a**) the SUV-H EAT volume, (**c**) serum LDH levels, or (**d**) BMI between complete and partial responders before treatment, with *p* values of 0.86, 0.18, and 0.64, respectively. (**b**) Greater initial SUV-H OAT volume is observed in complete responders than in partial responders (*p* = 0.02). * Differences are significant at the 0.05 level (two-tailed). SUV-H EAT, high standardized uptake value epicardial adipose tissue; SUV-H OAT, high standardized uptake value orbital adipose tissue; LDH, lactate dehydrogenase; BMI, body mass index; CR, complete response; PR, partial response.

**Figure 8 ijms-24-02158-f008:**
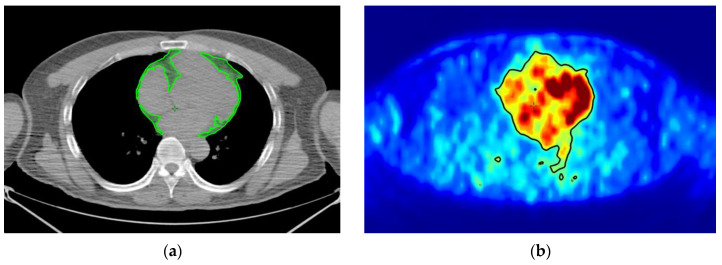
Image analyses of SUV-H EAT and SUV-H OAT. (**a**) EAT, with a specific HU between −190 and −10, is shown in green in the CT images. (**b**) SUV ≥ 1.5 is defined as a high SUV. Regions of SUV ≥ 1.5 are contoured with black lines in the PET images. (**c**) The intersection of (**a**,**b**) is defined as SUV-H EAT and is demonstrated in blue. (**d**) OAT with a HU between −190 and −10 is outlined in pink. (**e**) Regions of SUV ≥ 1.5 are contoured with black lines. (**f**) The intersection of (**d**,**e**) is defined as SUV-H OAT and is outlined in orange. SUV-H EAT, high standardized uptake value epicardial adipose tissue; SUV-H OAT, high standardized uptake value orbital adipose tissue; PET/CT, positron emission tomography/computed tomography; HU, Hounsfield unit.

**Table 1 ijms-24-02158-t001:** Baseline characteristics and IHC profiles of the 20 patients.

Characteristics		Responders (*n* = 17)	Non-Responders (*n* = 3)
Diagnosis			
FL		12	1
DLBCL		5	2
R-CHOP cycles			
4		3	1
6		7	1
8		7	1
PS			
0		13	2
1		4	1
Gender			
Male		10	2
Female		7	1
Median age (years)		61 (range, 30–76)	67 (range, 67–75)
Stage			
I		1	0
II		3	1
III		4	0
IV		9	2
Risk group ^1^			
Low		3	0
Intermediate		10	3
High		4	0
LDH (U/L) ^2^		277.9 ± 50.3	377.7 ± 97.2
BMI (kg/m^2^) ^2^		24.7 ± 1.1	24.9 ± 3.1
IHC profiles			
CD20	+	17	3
	−	0	0
CD10	+	10	2
	−	7	1
Bcl-6	+	9	3
	−	8	0
Bcl-2	+	11	3
	−	6	0

^1^ FL and DLBCL are risk-stratified by FLIPI and IPI, respectively. Five patients with DLBCL were responders. There were 2, 1, and 2 patients in the low, low-intermediate, and high-intermediate groups, respectively. Two patients with DLBCL were non-responders: one in the low-intermediate risk group and the other one in the high-intermediate risk group. ^2^ Baseline LDH levels and BMI are expressed as mean ± standard deviation. Abbreviations: IHC, immunohistochemical; FL, follicular lymphoma; DLBCL, diffuse large B-cell lymphoma; R-CHOP, rituximab, cyclophosphamide, doxorubicin, vincristine, and prednisone; PS, performance status; LDH, lactate dehydrogenase; BMI, body mass index; FLIPI, follicular lymphoma international prognostic index; IPI, international prognostic index.

**Table 2 ijms-24-02158-t002:** Delta SUV-H EAT and SUV-H OAT volumes, serum LDH levels, and BMI with different treatment responses.

	Responders (*n* = 17)		Non-Responders (*n* = 3)		
	M	SD	M	SD	*p*
SUV-H EAT (mL)	14.8	6.5	−8.6	6.5	0.03 ^1^
SUV-H OAT (mL)	3.1	0.8	−1.0	0.4	0.001 ^2^
LDH (U/L)	−11.4	66.5	398.7	151.2	0.03 ^1^
BMI (kg/m^2^)	−0.7	0.7	−2.3	1.6	0.36

^1^ Differences are significant at the 0.05 level (two-tailed). ^2^ Differences are significant at the 0.01 level (two-tailed). Abbreviations: SUV-H EAT, high standardized uptake value epicardial adipose tissue; SUV-H OAT, high standardized uptake value orbital adipose tissue; LDH, lactate dehydrogenase; BMI, body mass index; M, mean; SD, standard deviation.

**Table 3 ijms-24-02158-t003:** Delta SUV-H EAT and SUV-H OAT volumes, serum LDH levels, and BMI in complete and partial responders.

	Complete Responders (*n* = 10)		Partial Responders (*n* = 7)		
	M	SD	M	SD	*p*
SUV-H EAT (mL)	24.2	10.1	1.3	1.1	0.04 ^1^
SUV-H OAT (mL)	3.5	0.8	2.4	1.7	0.55
LDH (U/L)	−23.0	49.2	5.3	153.2	0.84
BMI (kg/m^2^)	−0.2	0.6	−1.4	1.3	0.37

^1^ Differences are significant at the 0.05 level (two-tailed). Abbreviations: SUV-H EAT, high standardized uptake value epicardial adipose tissue; SUV-H OAT, high standardized uptake value orbital adipose tissue; LDH, lactate dehydrogenase; BMI, body mass index; M, mean; SD, standard deviation.

**Table 4 ijms-24-02158-t004:** Pearson’s correlation analysis of delta SUV-H EAT and OAT volumes, LDH levels, and BMI.

	SUV-H EAT	SUV-H OAT	LDH	BMI
SUV-H EAT	1.00	0.50 ^1^	−0.06	0.23
SUV-H OAT		1.00	−0.70 ^2^	0.23
LDH			1.00	−0.01
BMI				1.00

^1^ Correlation is significant at the 0.05 level (two-tailed). ^2^ Correlation is significant at the 0.01 level (two-tailed). Abbreviations: SUV-H EAT, high standardized uptake value epicardial adipose tissue; SUV-H OAT, high standardized uptake value orbital adipose tissue; LDH, lactate dehydrogenase; BMI, body mass index.

**Table 5 ijms-24-02158-t005:** Pearson’s correlation analysis of initial SUV-H EAT and OAT volumes, LDH levels, and BMI.

	SUV-H EAT	SUV-H OAT	LDH	BMI
SUV-H EAT	1.00	0.07	0.38	−0.11
SUV-H OAT		1.00	−0.55 ^1^	0.06
LDH			1.00	−0.11
BMI				1.00

^1^ Correlation is significant at the 0.05 level (two-tailed). Abbreviations: SUV-H EAT, high standardized uptake value epicardial adipose tissue; SUV-H OAT, high standardized uptake value orbital adipose tissue; LDH, lactate dehydrogenase; BMI, body mass index.

**Table 6 ijms-24-02158-t006:** Logistic regression analysis for possible indicators of responders.

Indicators		OR	95% CI	*p*
Delta SUV-H EAT volume (mL)	≥0	1		
	<0	32.00	1.39–737.46	0.03 ^1^
Delta SUV-H OAT volume (mL)	≥−0.2	1		
	<−0.2	8.00	0.35–184.36	0.19
Delta LDH (U/L)	<200	1		
	≥200	32.00	1.39–737.46	0.03 ^1^
Delta BMI (kg/m^2^)	≥−0.2	1		
	<−0.2	2.86	0.22–37.99	0.43
Gender	Female	1		
	Male	1.40	0.11–18.62	0.79
Performance status	ECOG = 0 ^2^	1		
	ECOG = 1 ^2^	1.63	0.12–22.98	0.72
Stage	I and II	1		
	III and IV	1.63	0.12–22.98	0.72
Age (y)	<70	1		
	≥70	8.00	0.35–184.36	0.19

^1^ Correlation is significant at the 0.05 level (two-tailed). ^2^ The performance status is graded with the ECOG score, in which grade 0 indicates fully active and grade 1 indicates able to perform light work. Abbreviations: OR, odds ratio; CI, confidence interval; SUV-H EAT, high standardized uptake value epicardial adipose tissue; SUV-H OAT, high standardized uptake value orbital adipose tissue; LDH, lactate dehydrogenase; BMI, body mass index; ECOG, Eastern Cooperative Oncology Group.

## Data Availability

Not applicable.

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
