# Peer review of "Treatment Resulting Changes in Volumes of High-18F-FDG-Uptake Adipose Tissues over Orbit and Epicardium Correlate with Treatment Response for Non-Hodgkin’s Lymphoma"

_ijms, 2023, doi:10.3390/ijms24032158_

Round 1

Reviewer 1 Report (Previous Reviewer 1)

The manuscript focusing on brown adipose tissue to correlate with the therapeutic response for Non-Hodgkin’s Lymphoma is novel study. The main strength of this manuscript is the final part of discussion, which clearly demonstrated the limitations of the study. The authors also analyzed the logistic regression in the modified version. Thanks again for your wonderful effort to make an interesting manuscript.

Author Response

Thank you for reviewing our manuscript and for providing supportive comments. We appreciate the effort you made to improve this manuscript and are grateful for your insightful comments.

Reviewer 2 Report (New Reviewer)

In this manuscript, Huang et al. explored practical biomarkers for non-Hodgkin’s lymphoma by analyzing the metabolic activity of adipose tissue using 18F-FDG-PET/CT. 18F-FDG PET/CT remains the gold standard for detecting brown adipose tissue in humans and a previous study found no association between BAT and lymphoma’s metabolic activity (doi: 10.1038/s41598-020-78419-7). The authors pre-divided patients treated with R-CHOP (standard treatment) into responders (n=17) and non-responders (n=3) and observed a greater change in high-SUV of epicardial and orbital adipose tissue volume among responders. Overall, I think it’s an interesting observation, however, several things need revision:

1)     The authors need to determine the overall BAT status of all patients. How many of the patients were BAT-positive vs. BAT-negative pre-treatment?

2)     Can the authors based on 1) determine which patients will respond to R-CHOP treatment?

3)     High SUV is indicative of a more metabolically active adipose tissue which seems counterintuitive to the fact cancer cachexia often is associated with the increased metabolic activity of the adipose tissue and more brown adipose tissue. How do the authors explain that high SUV of the epicardial and orbital adipose tissue in this specific case is linked to a favorable response?

4)     A much more in-depth description of the scanning protocol is needed. Though 18F-FDG PET/CT remains the gold standard for detecting BAT in humans, scanning protocols in cancer patients attempt to suppress BAT activation by keeping patients in thermoneutral conditions, resulting in sub-maximal activation of BAT and potential underestimation of its prevalence and activity. Similarly, patients with systemic insulin resistance may have lower 18F-FDG uptake, resulting in a decreased likelihood of detecting BAT in this population. The authors need to specifically describe the condition the patients were subjected to – it is well-known that many independent parameters are associated with BAT activity (e.g., season of PET scan, the temperature in the room, beta-receptor therapy, metabolic diseases, cancer cachexia, age BMI, sex, etc.)  

5)     Given the little n of patients, it would make the most sense to plot the data in such a way that each individual data point illustrates individual patient-specific local changes from pre- to post-treatment.  

Author Response

Thank you for reviewing our manuscript and for providing supportive comments. We appreciate the effort you made to improve this manuscript and are grateful for your insightful comments. We have made point-by-point responses to the comments of each reviewer.

Comments of reviewer

  1. The authors need to determine the overall BAT status of all patients. How many of the patients were BAT-positive vs. BAT-negative pre-treatment?
  2. Can the authors based on 1) determine which patients will respond to R-CHOP treatment?
  3. High SUV is indicative of a more metabolically active adipose tissue which seems counterintuitive to the fact cancer cachexia often is associated with the increased metabolic activity of the adipose tissue and more brown adipose tissue. How do the authors explain that high SUV of the epicardial and orbital adipose tissue in this specific case is linked to a favorable response?
  4. A much more in-depth description of the scanning protocol is needed. Though 18F-FDG PET/CT remains the gold standard for detecting BAT in humans, scanning protocols in cancer patients attempt to suppress BAT activation by keeping patients in thermoneutral conditions, resulting in sub-maximal activation of BAT and potential underestimation of its prevalence and activity. Similarly, patients with systemic insulin resistance may have lower 18F-FDG uptake, resulting in a decreased likelihood of detecting BAT in this population. The authors need to specifically describe the condition the patients were subjected to – it is well-known that many independent parameters are associated with BAT activity (e.g., season of PET scan, the temperature in the room, beta-receptor therapy, metabolic diseases, cancer cachexia, age BMI, sex, etc.)
  5. Given the little n of patients, it would make the most sense to plot the data in such a way that each individual data point illustrates individual patient-specific local changes from pre- to post-treatment.

Responses to reviewer

  1. Thanks for your great comment. This study did not evaluate the overall BAT status of all patients because the metabolic activity of BAT in PET/CT is easily affected by cancer and inflammation. EAT and OAT share similarities that are both rarely affected by malignancies, and remote from sites of cancer treatments. Therefore, we choose EAT and OAT for evaluation. The Discussion is revised as follows.

Previous version:

“Malignant orbital tumors are unusual. EAT and OAT share similarities that are both rarely affected by malignancies, and remote from sites of cancer treatments.”

Revised version:

“Malignant orbital tumors are unusual. This study did not evaluate the overall BAT status of all patients because the metabolic activity of BAT in PET/CT is easily affected by cancer and inflammation. EAT and OAT share similarities that are both rarely affected by malignancies, and remote from sites of cancer treatments. Therefore, we choose EAT and OAT for evaluation.”

  1. We’ve focused on the change of SUV-H EAT and SUV-H OAT volume after R-CHOP, not the initial status of BAT. As mentioned in Discussion , delta volume of SUV-H EAT and SUV-H OAT is the main issue discussed in this study, and the bias from individualized variation may be reduced by comparing changes, not only one time point. Greater changes in SUV-H EAT and SUV-H OAT volume after R-CHOP in responders than in non-responders are observed in our study. It may imply that patients with elevated SUV-H EAT and SUV-H OAT volume will respond to R-CHOP. However, this hypothesis needs more investigation. The 5. Conclusion is revised as follows.

Previous version:

“Greater changes in SUV-H EAT and SUV-H OAT volume, which may represent more browning in EAT and OAT after R-CHOP therapy in responders than in non-responders, indicate that BAT could be regarded as a favorable prognostic biomarker. This observation is supported by PET/CT scans and serum LDH profiles.”

Revised version:

“Greater changes in SUV-H EAT and SUV-H OAT volume, which may represent more browning in EAT and OAT after R-CHOP therapy in responders than in non-responders, indicate that BAT could be regarded as a favorable prognostic biomarker. This observation is supported by PET/CT scans and serum LDH profiles. It may imply that patients with elevated SUV-H EAT and SUV-H OAT volume will respond to R-CHOP, but this hypothesis requires more investigation.”

  1. Thanks for your practical comment. High SUV is indicative of a more metabolically active adipose tissue which seems related to cancer cachexia in the literatures. In our study, no cachexia is observed during the treatment and follow-up period, and no significant BMI changes are reported. Therefore, the relationship of metabolically active adipose tissue and cancer cachexia could not be drawn in this study. Several paragraphs are added in Materials and Methods for explanation as follows.

“No cancer cachexia was observed in these patients, and no significant BMI changes were reported during the treatment and follow-up period. The clinical characteristics of each patient including gender, age, and BMI were recorded in Table 1.”

We previously demonstrated that EAT might serve as a biomarker of survival outcomes in patients with esophageal cancer receiving neoadjuvant chemoradiation therapy. OAT was recently considered to be related to Graves' orbitopathy, an autoimmune-driven manifestation of Graves' disease. Therefore, EAT and OAT were chosen for evaluation in this study. R-CHOP may influence EAT and OAT to induce local or systemic inflammation or immune response, which could be related to treatment response of NHL. Several paragraphs are added in 3. Discussion for explanation as follows.

“EAT and OAT might play a role in the immune system. We previously demonstrated that EAT may serve as a biomarker of survival outcomes in patients with esophageal cancer receiving neoadjuvant chemoradiation therapy [38]. EAT is composed of adipocytes, nerve cells, inflammatory cells (mainly macrophages and mast cells), stromal cells, vascular cells, and immune cells [21,39], and functions as BAT with the expression of UCP1, brown adipocyte differentiation transcription factor PR-domain-missing 16, and peroxisome-proliferator-activated receptor γ co-activator-1α [40]. In patients with coronary artery disease, local expression of chemokine (monocyte chemotactic protein 1 (MCP1)) and inflammatory cytokines (IL-1β, IL-6, and TNF-α) was observed with significant changes in MCP1, IL-1β, IL-6, and TNF-α mRNA and protein in the epicardial adipose stores [41,42]. Philipp et al. have developed a mouse model to study Graves' orbitopathy (GO), and found early infiltration of macrophages in the orbital region, induction of anti-thyroid stimulating hormone receptor antibodies, aggregation of CD8+ T cells, and BAT increase during GO onset [39]. R-CHOP may influence EAT and OAT to induce local or systemic inflammation or immune response, which could be related to treatment response of NHL.”

  1. Thanks for the opportunity to let us detail the PET/CT scanning protocol. The Materials and Methods is revised as follows.

Previous version:

“Whole body PET/CT (GE Discovery, GE Healthcare, Milwaukee, WI, USA) was performed after blood glucose measurement. Patients were asked to fast for at least 6 hours before the examination, and 18F-FDG was injected only if blood glucose was less than 200 mg/dL. CT was obtained in mid-expiratory phase without intravenous contrast (slice thickness, 5 mm; tube voltage, 120 kVp; tube current, 40 mAs; field of view, 50 cm), and PET images were reconstructed. The CT used in this study was examined according to the American Association of Physicists in Medicine (AAPM) and American College of Radiology (ACR) guidelines (AAPM report #74 and #96 and ACR CT QC manual), and standard quality assurance measures were performed.”

Revised version:

“Whole body PET/CT (GE Discovery, GE Healthcare, Milwaukee, WI, USA) was performed after blood glucose measurement with a constant temperature of 22 °C regardless of the season. Patients were asked to fast for at least 6 hours before the examination, and 18F-FDG was injected only if blood glucose was less than 200 mg/dL. Then, the patients rested during the uptake time for 60 minutes in an air-conditioned waiting room with a constant temperature of 22 °C. CT was obtained in mid-expiratory phase without intravenous contrast (slice thickness, 5 mm; tube voltage, 120 kVp; tube current, 40 mAs; field of view, 50 cm), and PET images were reconstructed. The CT used in this study was examined according to the American Association of Physicists in Medicine (AAPM) and American College of Radiology (ACR) guidelines (AAPM report #74 and #96 and ACR CT QC manual), and standard quality assurance measures were performed. All included patients had no metabolic diseases as hypertension and diabetes mellitus. No patients received therapies related to beta-adrenergic receptor. No cancer cachexia was observed in these patients, and no significant BMI changes were reported during the treatment and follow-up period. The clinical characteristics of each patient including gender, age, and BMI were recorded in Table 1.”

  1. Thanks for your suggestion. We’ve provided Figure 2 and Figure 3 to show the changes of SUV-H EAT, SUV-H OAT volume, serum LDH levels, and BMI in different treatment responses, and found that SUV-H EAT and SUV-H OAT volume significantly increased in responders after R-CHOP. Therefore, we’ve plotted each individual data point on Supplementary Figure S1 and S2 to illustrate the patient-specific local changes in SUV-H EAT and SUV-H OAT volume from pre- to post-treatment.

Reviewer 3 Report (New Reviewer)

Thank you for the opportunity to review this interesting research regarding the potential role of R-CHO therapy in the 'browning' of EAT and  OAT in NHL patients, verified by PET/CT scans.

To improve the manuscript, I recommend:

-          In lines 124-142, almost all data presented can be found in table 1. There is no need to repeat this. Where were CD20, CD10, Bcl-2, and Bcl-6 positive immunoexpression observed?

-          Kindly explain all the abbreviations used in the Abstract and the main text of the manuscript

-          In lines 63, 101, and 105, kindly add references related to the data presented.

Author Response

Thank you for reviewing our manuscript and for providing supportive comments. We appreciate the effort you made to improve this manuscript and are grateful for your insightful comments. We have made point-by-point responses to the comments of each reviewer.

Comments of reviewer

  1. In lines 124-142, almost all data presented can be found in table 1. There is no need to repeat this. Where were CD20, CD10, Bcl-2, and Bcl-6 positive immunoexpression observed?
  2. Kindly explain all the abbreviations used in the Abstract and the main text of the manuscript.
  3. In lines 63, 101, and 105, kindly add references related to the data presented.

Responses to reviewer

  1. Thanks for your great comment. We have cut down the repeated paragraphs in lines 124-142. The CD20, CD10, Bcl-2, and Bcl-6 positive immunoexpression were observed in lymph node tissue removed during biopsy. The 1. Patient characteristics is revised as follows.

Previous version:

“Table 1 summarizes the baseline characteristics and immunohistochemical (IHC) profiles of the 20 patients. Of the 20 patients, 17 (85%) were responders, and 3 (15%) were non-responders. Complete response (CR) was noted in 10 patients, partial response (PR) in 7 patients, and progressive disease (PD) in 3 patients according to the post-treatment PET/CT scans. Thirteen (65%) patients had FL, of which 12 were responders and 1 was non-responder. Seven (35%) patients were diagnosed with germinal center B-cell-like DLBCL, of which 5 were responders and 2 were non-responders. All patients completed the R-CHOP regimens as scheduled: 4, 6, and 8 cycles in 4, 8, and 8 patients, respectively. All patients had a good performance status with an Eastern Cooperative Oncology Group performance status score of 0–1. Twelve (60%) patients were men, and 8 (40%) were women. The median age at diagnosis was 63 years (range, 30–76 years). In responders, 1, 3, 4, and 9 patients were diagnosed with clinical stage I, II, III, and IV disease, respectively; while 1 and 2 patients were diagnosed with clinical stage II and IV disease in non-responders, respectively. Responders were classified as 3, 10, and 4 patients in low, intermediate, and high-risk groups, respectively; while all non-responders were in intermediate-risk group. The baseline LDH levels were 277.9 ± 50.3 U/L and 377.7 ± 97.2 U/L in responders and non-responders, respectively. The baseline BMI was 24.7 ± 1.1 kg/m2 and 24.9 ± 3.1 kg/m2 in responders and non-responders, respectively. All patients were CD20-positive in IHC profiles.”

Revised version:

“Table 1 summarizes the baseline characteristics and immunohistochemical (IHC) profiles of the 20 patients. Of the 20 patients, complete response (CR) was noted in 10 patients, partial response (PR) in 7 patients, and progressive disease (PD) in 3 patients according to the post-treatment PET/CT scans. IHC analysis was performed on lymph node tissue removed during biopsy. All patients were CD20-positive in IHC profiles.”

  1. Thanks for the opportunity to let us explain the abbreviations. The abbreviations are defined the first time they appear in the abstract and main text. When defined for the first time, the abbreviations are added in parentheses after the written-out form. The Abstract is revised as follows with abbreviations added.

Previous version:

“Background: R-CHOP is the standard treatment for non-Hodgkin’s lymphoma. Brown adipose tissue possesses anti-cancer potential. This study aims to explore practical biomarkers for non-Hodgkin’s lymphoma by analyzing the metabolic activity of adipose tissue. Methods: Twenty patients who received R-CHOP for non-Hodgkin’s lymphoma were reviewed. PET/CT images, LDH levels, and BMI before and after treatment were collected. Regions with high SUV in epicardial and orbital adipose tissue were selected and analyzed by a PET/CT viewer.”

Revised version:

“Background: Rituximab, cyclophosphamide, doxorubicin, vincristine, and prednisone (R-CHOP) is the standard treatment for non-Hodgkin’s lymphoma. Brown adipose tissue possesses anti-cancer potential. This study aims to explore practical biomarkers for non-Hodgkin’s lymphoma by analyzing the metabolic activity of adipose tissue. Methods: Twenty patients who received R-CHOP for non-Hodgkin’s lymphoma were reviewed. Positron emission tomography/computed tomography (PET/CT) images, lactate dehydrogenase (LDH) levels, and body mass index (BMI) before and after treatment were collected. Regions with high standardized uptake value (SUV) in epicardial and orbital adipose tissue were selected and analyzed by a PET/CT viewer.”

  1. Thanks for your suggestion. The references related to the data presented in lines 63, 101, and 105 are added.

Previous version (line 63):

“Non-Hodgkin’s lymphoma (NHL) is the tenth most common cancer and eleventh leading cause of cancer-related deaths worldwide.”

Revised version (line 63):

“Non-Hodgkin’s lymphoma (NHL) is the tenth most common cancer and eleventh leading cause of cancer-related deaths worldwide [1].”

Previous version (line 101):

“WAT, when exposed to certain stimuli (e.g., cold, exercise, or adrenergic receptor activation), may undergo morphologic and functional changes to transform into BAT.”

Revised version (line 101):

“WAT, when exposed to certain stimuli (e.g., cold, exercise, or adrenergic receptor activation), may undergo morphologic and functional changes to transform into BAT [15].”

Previous version (line 105):

“In recent years, several imaging strategies have been used in studies related to BAT [24]. To date, positron emission tomography/computed tomography (PET/CT) using the tracer 18F-fluorodeoxyglucose (18F-FDG) has been the most commonly used strategy.”

Revised version (line 105):

“In recent years, several imaging strategies have been used in studies related to BAT [24]. To date, positron emission tomography/computed tomography (PET/CT) using the tracer 18F-fluorodeoxyglucose (18F-FDG) has been the most commonly used strategy [24,25].”

This manuscript is a resubmission of an earlier submission. The following is a list of the peer review reports and author responses from that submission.

Round 1

Reviewer 1 Report

This quite appreciable manuscript has focused on brown adipose tissue to correlate with the therapeutic response for Non-Hodgkin’s Lymphoma. The main strength of this manuscript is the final part of discussion, which clearly demonstrated the limitations of the study. Although there are some issues, which need to address:

1)     Please revise 2nd and 3rd sentences: “It is an……………molecules that can fuel cancer development” on line 81-84. As white adipose tissue is not metabolically active (but beige and brown adipose tissue), and the mentioned reference (8) didn’t address “white”, kindly rephrase the sentences. These sentences are challenging the next stated sentences in the manuscript.

2)     “Patient characteristics” are not clear on result section. Please rewrite this part carefully and check table 1.

3)     Please write in details of table 4 and 5 on “2.5. Correlation analysis” section.     

Thanks again for your wonderful effort to make an interesting manuscript. 

Author Response

Thanks for the suggestion for this article. We have made some point-by-point responses to the comments of reviewer.

Comments of reviewer

  1. Please revise 2nd and 3rd sentences: “It is an……………molecules that can fuel cancer development” on line 81-84. As white adipose tissue is not metabolically active (but beige and brown adipose tissue), and the mentioned reference (8) didn’t address “white”, kindly rephrase the sentences. These sentences are challenging the next stated sentences in the manuscript.
  2. “Patient characteristics” are not clear on result section. Please rewrite this part carefully and check table 1.
  3. Please write in details of table 4 and 5 on “2.5. Correlation analysis” section.

Responses to reviewer

  1. Thanks for your great comment. The white adipose tissue is not metabolically active. The Introduction is revised as follows.

Previous version:

“White adipose tissue (WAT) is the most abundant tissue in the human body. It is an energy source and an active metabolic organ secreting several adipokines, such as leptin and adiponectin. WAT can store and release energy in the form of lipids, and secrete a variety of inflammatory molecules that can fuel cancer development.”

Revised version:

“White adipose tissue (WAT) is the most abundant tissue in the human body. It is an energy source, which can store and release energy in the form of lipids.”

  1. Thanks for the opportunity to let us detail the patient characteristics in results. The 1. Patient characteristics is revised as follows.

Previous version:

“Table 1 summarizes the baseline characteristics and immunohistochemical (IHC) profiles of the 20 patients. Of the 20 patients, 12 (60%) were men. Thirteen patients (65%) had FL and the rest were diagnosed with germinal center B-cell-like DLBCL. The median age at diagnosis was 63 years (range, 30–76 years). Patients with FL were classified as 1, 8, and 4 patients in low, intermediate, and high-risk groups by FLIPI, respectively; while patients with DLBCL were classified as 2, 2, and 3 patients in low, low-intermediate, and high-intermediate groups by IPI. All patients completed the R-CHOP regimens as scheduled: 4, 6, and 8 cycles in 4, 8, and 8 patients, respectively. Complete response (CR) was noted in 10 patients, partial response (PR) in 7 patients, and progressive disease (PD) in 3 patients according to the post-treatment PET/CT scans.”

Revised version:

“Table 1 summarizes the baseline characteristics and immunohistochemical (IHC) profiles of the 20 patients. Of the 20 patients, 17 (85%) were responders, and 3 (15%) were non-responders. Complete response (CR) was noted in 10 patients, partial response (PR) in 7 patients, and progressive disease (PD) in 3 patients according to the post-treatment PET/CT scans. Thirteen (65%) patients had FL, of which 12 were responders and 1 was non-responder. Seven (35%) patients were diagnosed with germinal center B-cell-like DLBCL, of which 5 were responders and 2 were non-responders. All patients completed the R-CHOP regimens as scheduled: 4, 6, and 8 cycles in 4, 8, and 8 patients, respectively. All patients had a good performance status with an Eastern Cooperative Oncology Group performance status score of 0–1. Twelve (60%) patients were men, and 8 (40%) were women. The median age at diagnosis was 63 years (range, 30–76 years). In responders, 1, 3, 4, and 9 patients were diagnosed with clinical stage I, II, III, and IV disease, respectively; while 1 and 2 patients were diagnosed with clinical stage II and IV disease in non-responders, respectively. Responders were classified as 3, 10, and 4 patients in low, intermediate, and high-risk groups, respectively; while all non-responders were in intermediate-risk group. The baseline LDH levels were 277.9 ± 50.3 U/L and 377.7 ± 97.2 U/L in responders and non-responders, respectively. The baseline BMI was 24.7 ± 1.1 kg/m2 and 24.9 ± 3.1 kg/m2 in responders and non-responders, respectively. All patients were CD20-positive in IHC profiles.”

  1. Thanks for the opportunity to let us detail the correlation analysis in results. The 5. Correlation analysis is revised as follows.

Previous version:

“Pearson's correlation analyses were performed, and correlation coefficients (r) were recorded. The results showed a strong correlation between delta SUV-H OAT volume and delta LDH levels (r = −0.70) (Table 4), and a negative correlation between initial SUV-H OAT volume and initial LDH levels (r = −0.55) (Table 5).”

Revised version:

“Pearson's correlation analyses were performed to analyze the correlations among delta and initial SUV-H EAT, SUV-H OAT volume, LDH levels, and BMI, respectively. The correlation coefficients (r) were recorded in Table 4 and Table 5. A strong negative correlation of delta SUV-H OAT volume and delta LDH levels was detected (r = −0.70, p = 0.002), and a significant positive correlation of delta SUV-H EAT and OAT volume was noted (r = 0.50, p = 0.03). A significant negative correlation of initial SUV-H OAT volume and initial LDH levels was observed (r = −0.55, p = 0.02). No other significant correlations were shown in the correlation analyses.”

Reviewer 2 Report

This work focuses on obtaining new biomarkers that allow evaluating the response of patients with non-Hodgkin's lymphoma (NHL) in treatment with Rituximab, cyclosphosphamide, doxorubicin, vincristine and prednisone (R-CHOP). It is proposed that the analysis of epicardial as well as orbital adipose tissue could be considered biomarkers of this type. To reach this conclusion, a retrospective study was carried out with 20 patients, registering several variables aimed at evaluating their oncological process, before and after treatment.

Some points need to be clarified:

- Page 3, line 125: One of the recognized limitations of this work (page 13, line 323) is the number of patients included in the study. This fact could be generating some bias in the results. For example, in some cases, such as risk stratification, there are no non-responders in each of the risk groups. It is recommended to include more patients in the study.

- Page 11, line 259: It is recommended to specifically include the p value, at least in those cases where it is significant. It is only reflected if it was less than 0.05 or 0.01.

- Page 15, line 386: This section describes the statistical methodology used. On the one hand, it is reflected that comparisons of two groups were made using t-tests, either with the observations before and after treatment (paired t-tests), or with the groups that responded or not to the treatment, as well as complete and partial responders (two-sample t-tests). On the other hand, it is reflected that delta values ​​are presented that represent the difference between two values. The statistical analysis carried out is not entirely correct. Having performed t-tests, not all the variables collected were considered. For example, it is useless to assess the volume of epicardial adipose tissue in relation to the response or not to treatment, if one does not consider the risk group to which the patient belongs, the stage of the oncological process, age or sex. In this way, a global vision of the results is not obtained. For this, it is recommended to carry out a multivariate analysis in greater depth to address all the variables collected so that the results obtained have greater weight when interpreting them.

Best regards

Author Response

Thanks for the suggestion for this article. We have made some point-by-point responses to the comments of reviewer.

Comments of reviewer

  1. Page 3, line 125: One of the recognized limitations of this work (page 13, line 323) is the number of patients included in the study. This fact could be generating some bias in the results. For example, in some cases, such as risk stratification, there are no non-responders in each of the risk groups. It is recommended to include more patients in the study.
  2. Page 11, line 259: It is recommended to specifically include the p value, at least in those cases where it is significant. It is only reflected if it was less than 0.05 or 0.01.
  3. Page 15, line 386: This section describes the statistical methodology used. On the one hand, it is reflected that comparisons of two groups were made using t-tests, either with the observations before and after treatment (paired t-tests), or with the groups that responded or not to the treatment, as well as complete and partial responders (two-sample t-tests). On the other hand, it is reflected that delta values ​​are presented that represent the difference between two values. The statistical analysis carried out is not entirely correct. Having performed t-tests, not all the variables collected were considered. For example, it is useless to assess the volume of epicardial adipose tissue in relation to the response or not to treatment, if one does not consider the risk group to which the patient belongs, the stage of the oncological process, age or sex. In this way, a global vision of the results is not obtained. For this, it is recommended to carry out a multivariate analysis in greater depth to address all the variables collected so that the results obtained have greater weight when interpreting them.

Responses to reviewer

  1. Thanks for your great comment. The small number of patients included in the study is a recognized limitation of this work. Our preliminary findings of greater changes in SUV-H EAT and OAT volumes among responders have provided critical information for further research. We would like to recommend including more patients in the future prospective study. The Discussion is revised as follows for further explanation.

Previous version:

“Second, the number of patients in our retrospective study with PET/CT and serum LDH profiles was too small to draw a conclusion.”

Revised version:

“Second, the number of patients in our retrospective study with PET/CT and serum LDH profiles was too small to draw a firm conclusion. This fact could be generating some bias in the results. For example, in some cases, such as risk stratification, there were no non-responders in each of the risk groups. Our preliminary findings of greater changes in SUV-H EAT and OAT volumes among responders have provided critical information for conducting further prospective study with more enrolled patients.”

  1. Thanks for your suggestion. The p value is added in those cases where it is significant. The 5. Correlation analysis is revised as follows.

Previous version:

“Pearson's correlation analyses were performed, and correlation coefficients (r) were recorded. The results showed a strong correlation between delta SUV-H OAT volume and delta LDH levels (r = −0.70) (Table 4), and a negative correlation between initial SUV-H OAT volume and initial LDH levels (r = −0.55) (Table 5).”

Revised version:

“Pearson's correlation analyses were performed to analyze the correlations among delta and initial SUV-H EAT, SUV-H OAT volume, LDH levels, and BMI, respectively. The correlation coefficients (r) were recorded in Table 4 and Table 5. A strong negative correlation of delta SUV-H OAT volume and delta LDH levels was detected (r = −0.70, p = 0.002), and a significant positive correlation of delta SUV-H EAT and OAT volume was noted (r = 0.50, p = 0.03). A significant negative correlation of initial SUV-H OAT volume and initial LDH levels was observed (r = −0.55, p = 0.02). No other significant correlations were shown in the correlation analyses.”

  1. Thanks for your practical comment. A multivariate analysis is recommended to address all the variables, and the results may have greater weight when interpreting them. However, there are no independent predictors in multivariate analysis, which may be related to only 3 non-responders in this study. If more patients included, more information may be shown in the multivariate analysis. Our preliminary findings of greater changes in SUV-H EAT and OAT volumes among responders have provided critical information for further research. A larger prospective study and experimental animal models to validate our findings are warranted.

Round 2

Reviewer 2 Report

Despite being some initial observations, and that these could have greater relevance when the study sample is increased, these results are preliminary, conclusions of great weight cannot be established with very small samples. It is recommended that the records be increased in order to carry out a correct statistical study.

Best regards